# Friction and Temperature Behavior of Lubricated Thermoplastic Polymer Contacts

**Stefan Reitschuster \*, Enzo Maier, Thomas Lohner and Karsten Stahl**

Gear Research Centre (FZG), Technical University of Munich, Boltzmannstraße 15,
D-85748 Garching near Munich, Germany; e.maier@fzg.mw.tum.de (E.M.); lohner@fzg.mw.tum.de (T.L.);
stahl@fzg.mw.tum.de (K.S.)
**\*** Correspondence: reitschuster@fzg.mw.tum.de

**Abstract:** This work focuses on the friction and temperature behavior of thermo-elastohydrodynamically lubricated (TEHL) contacts under rolling-sliding conditions. For this purpose, a twin-disk test rig is used with a hybrid setup of plain and fiber-reinforced polyamide (PA) 66 and polyetheretherketone (PEEK) disks paired with case-hardened steel disks and three different lubricants. Experimental investigations include various lubrication regimes by varying sum velocity and oil temperature as well as load and slip ratio. The measured friction in thermoplastic TEHL contacts is particularly very low in the area of high fluid load portion, which refers to the large deformation of the compliant polymer surface. Newtonian flow behavior mainly determines fluid friction. The low thermal effusivity of polymers insulates the contact and can further reduce the effective lubricant viscosity, and thus the fluid friction. For low sum velocities, solid friction influences the tribological behavior depending on the solid load portion. Although the interfacial contact friction is comparably small, material damping strongly contributes to power losses and increases bulk temperature, which in turn affects the TEHL contact. Thus, loading frequency and the resulting bulk temperature are identified as one of the main drivers of power losses and tribological behavior of lubricated thermoplastic polymer contacts.

**Keywords:** TEHL contact; polymers; friction; temperature; solid losses

## 1. Introduction

Thermoplastic polymers are increasingly introduced to machine elements owing to low density and high noise-damping capacity. Additionally, injection molding allows for the cost-effective manufacturing of thermoplastic components [1,2]. In contrast, the moderate mechanical properties and their strong temperature dependency limit their use to lowly-stressed applications. Highly-stressed machine elements making of thermoplastics are often lubricated to dissipate frictional losses, while at the same time reducing friction and wear. As a result of this, a significant increase in the transmittable power can be achieved, as exemplarily shown by Hasl et al. [3] and Illenberger et al. [4] for polymer gears.

Depending on the solid's stiffness and the lubricant's viscosity in lubricated contacts, a thermo-elastohydrodynamically lubricated (TEHL) contact is formed. Soft TEHL contacts are generally defined by deformations of the rolling-sliding elements greater than the thickness of the lubricant film [5]. A detailed characterization of the soft TEHL contact is given by Maier et al. [6]. The behavior of the lubricant is described as both, nearly isoviscous and Newtonian (e.g., [6,7]). In TEHL contacts with thermoplastic elements, the large surface deformation owing to low polymer stiffness causes a significant decrease in hydrodynamic contact pressure and an increase in the lubricant film thickness. The associated pressure-viscosity increase is small, which results in low fluid friction [7]. According

to Myers et al. [8] and Dearn et al. [9], the TEHL contact with either one or two thermoplastic elements (thermoplastic TEHL contact) takes an intermediate position between the elastic-isoviscous and elastic-piezoviscous regime.

For contacts that show elastic surface deformation and constant viscosity, Vicente et al. [7] and Hamrock and Dowson [10] developed approaches to calculate interfacial friction and lubricant film thickness, which were partially confirmed by Myant et al. [11] and Stupkiewicz et al. [12] in experimental investigations on a ball-on-disk tribometer and a mini-traction machine. Coefficients of friction of $\mu < 0.1$ were shown for thermoplastic TEHL contacts. Hence, coefficients of friction in the range of superlubricity are possible [13–15]. Vicente et al. [7] and Persson and Scaraggi [16] point out that both rolling and sliding fluid friction have to be considered in thermoplastic TEHL contacts. In mixed lubrication, solid interfacial friction due to contact of surface asperities becomes relevant. To date, much research on polymer tribology has focused on lowly-stressed conditions. However, the tribology of thermoplastic TEHL contacts under highly-stressed conditions is approached in medical research, such as in the field of joint prosthesis, for example [17].

Under highly-stressed conditions, the influence of the thermal and viscoelastic behavior of the polymer on the tribological system becomes increasingly important, particularly in connection with the temperature-dependent mechanical properties. The glass transition temperature is especially important in the context of thermoplastic TEHL contact because it characterizes a temperature range in which material behavior changes from a hard, glassy state to a softer, rubber-like state. The transition occurs in different temperature and frequency ranges depending on the thermoplastic material. The associated energy dissipation due to material damping in the thermoplastic solid characterizes the friction and temperature behavior of thermoplastic TEHL contacts, in addition to the interfacial friction [18,19]. These losses are also referred to as viscoelastic or hysteresis friction. According to Hook and Huang [20], the viscoelasticity of materials affects the lubricant film formation and thickness and successively influences the pressure distribution. Putignano et al. [18] discuss the relationship between temperature and load frequency as well as the influence on material behavior (time–temperature superposition) and show the possibility of transforming viscoelastic friction curves for different temperatures into a single master curve using a frequency shift coefficient. Moreover, Putignano et al. [21] manage to isolate and measure the viscoelastic friction in experimental investigations on a mini-traction machine. The viscoelastic friction determined in this way depends largely on the deformation velocity of the body. Snoeijer et al. [22] investigate soft lubricated contacts and provide a theory for calculating the lubricant film thickness in good agreement with numerical simulation by considering viscoelastic material behavior and corresponding deformations. Putignano und Dini [23] study the interaction of fluid and solid with viscoelastic deformations, focusing on the lubricant film thickness and pressure distribution. For low loading frequencies, only small deviations from the elastic material behavior are shown. With increasing frequency, changes and displacement of the contact shape and size are detected, which also influence the lubricant film thickness distribution. A detailed understanding of friction and temperature as well as their origin and relationship, for highly-stressed thermoplastic TEHL contacts is still missing.

This work investigates the friction and temperature behavior of different thermoplastics in TEHL line contact configuration under operating conditions relevant for gears. On the basis of the experimental results obtained on an FZG twin-disk test rig, the observed polymer-specific phenomena are discussed to achieve a more detailed understanding of the influences of thermoplastic material behavior on the tribological system. The following work is divided into three sections. First, the experimental setup is described. Second, the experimental results are presented and discussed. This includes, on the one hand, bulk temperature measurements under increasing load and, on the other hand, investigations into friction and temperature behavior under different conditions and slip ratios. Third, the main findings are summarized in the conclusion.

## 2. Experimental Setup

This section describes the experimental setup of the FZG twin-disk test rig as well as the operating conditions, materials, lubricants, and experimental procedure.

### 2.1. Twin-Disk Test Rig and Specimen

The following description of functionality is mainly based on Lohner et al. [24]. The two cylindrical disks (⌀ 80 mm) are press-fitted onto parallel shafts that are independently driven by two electric motors, which allow a continuous variation of velocity. A pneumatic air cylinder at the end of a pivot arm on which the lower disk is mounted applies a normal force $F_N$ to the disk contact. The upper disk is mounted in a skid, which is firmly connected to the frame via steel sheets. A load cell supports the sideways movement of the skid in such a way that the contact friction force $F_R$ is measured without noticeable displacement of the skid. A nozzle injects preheated and filtered oil at different temperatures into the inlet zone of the disk contact. An oil pump provides a sufficient average oil flow of $\dot{Q} = 1.5 \, l/min$ to prevent starved lubrication. As shown in Figure 1b, a Pt-100 sensor measures the bulk temperature of the polymer disk 4 mm below the surface. Normal force $F_N$, frictional force $F_R$, oil injection temperature $\vartheta_{oil}$, surface velocities $v_1$ and $v_2$ of the lower and upper disk, as well as the bulk temperature $\vartheta_M$ of the upper disk are continuously recorded. The coefficient of friction is determined by $\mu = F_R/F_N$. The sum velocity $v_\Sigma = v_1 + v_2$ is a measure of the hydrodynamic velocity, with $v_1 \geq v_2$. The ratio of sliding velocity $v_g = v_1 - v_2$ and the faster surface velocity $v_1$ is termed slip ratio $s = v_g/v_1$.

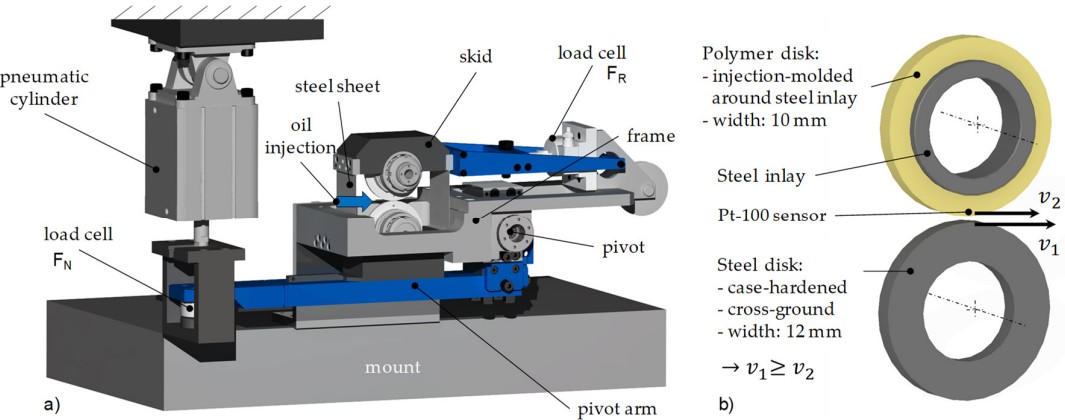

**Figure 1.** Layout of the FZG twin-disk test rig (**a**) and specimen design (**b**).

It should be noted that the test rig described here was originally designed for friction measurements of highly-stressed steel–steel contacts with high normal forces as well as high sliding velocities such as the analogy test rig for steel gear contacts. Owing to the measuring principle, only horizontally acting friction forces are measured, whereas loss torques are not registered by the friction load cell. Therefore, friction measurement is generally limited to interfacial friction in the contact. In polymer–steel pairings, external and material influencing factors can result in additional non-horizontal friction forces. For example, viscoelastic material behavior dampens surface deformation and normal stress. The associated energy dissipation in solids can be registered by a bulk temperature sensor. The measured coefficient of friction is referred to as μ* in the following.

Figure 1b illustrates the design of the specimens. The steel disks (16MnCr5) were case-hardened and axially ground to $Ra \approx 0.2 \, \mu m$. The polymer disks were injection molded around a perforated steel inlay for form fit and not post-treated. Four different polymer materials were considered: polyamide 66 (PA66) and polyetheretherketone (PEEK), both plain and 30 wt.-% short glass fiber reinforced (PA66+GF30 and PEEK+GF30). Owing to the flow direction of the melt during the injection molding process, the fibers of reinforced materials are oriented in a transverse direction to the

circumference. The material parameters given in Table 1 are standard values that apply to the dry state at ambient temperature. It must be considered that a change in temperature and ambient conditions can significantly influence these parameters. All specimens were stored under dry conditions and at room temperature until the start of the experiment.

**Table 1.** Material parameters at dry state at ambient temperature [2,25–34]. PA66, polyamide 66; PEEK, polyetheretherketone; GF, glass fiber.

|  | PA66 | PEEK | PA66 + GF30 | PEEK + GF30 | 16MnCr5 |
|---|---|---|---|---|---|
| Young's modulus E in MPa | 3100 | 3600 | 10,500 | 11,000 | 210,000 |
| Poisson number $\nu$ | 0.33 | 0.37 | 0.35 | 0.37 | 0.30 |
| Thermal conductivity $\lambda$ in $W/(mK)$ | 0.23 | 0.29 | 0.35 | 0.43 | 44.0 |
| Glass transition temperature $T_G$ in °C | 66 | 143 | 64 | 143 | - |

Figure 2 compares the surface structure and roughness of the considered polymer disks. The given surface impressions were recorded with an optical microscope with 5x magnification. For PEEK and PEEK+GF30 separate injection molding tools were used, which is reflected in an overall higher assessed surface quality. PA66 and PA66+GF30, on the other hand, were manufactured in one common injection mold. PA66+GF30 does not show a consistent surface structure over the disk width, probably as a result of the injection molding process. Depending on the injection mold, slightly different surface structures result, characterized by the negative surface imprint of the mold. The corresponding averaged profile roughness values shown in Figure 2 were determined using the tactile profile method. The measurement was carried out across the disk width direction with a measuring length of $L_t = 4.80\ mm$ and a cut-off wavelength of $\lambda_c = 0.80\ mm$ according to DIN EN ISO 4288 [35]. A total of at least 30 disks were measured for each material. Standard deviation for PEEK was less than 0.04 μm for Ra and 0.27 μm for Rz, for PEEK+GF30 less than 0.04 μm for Ra and 0.32 μm for Rz, for PA66 less than 0.06 μm for Ra and 0.31 μm for Rz, for PA66+GF30 less than 0.21 μm for Ra and 1.22 μm for Rz.

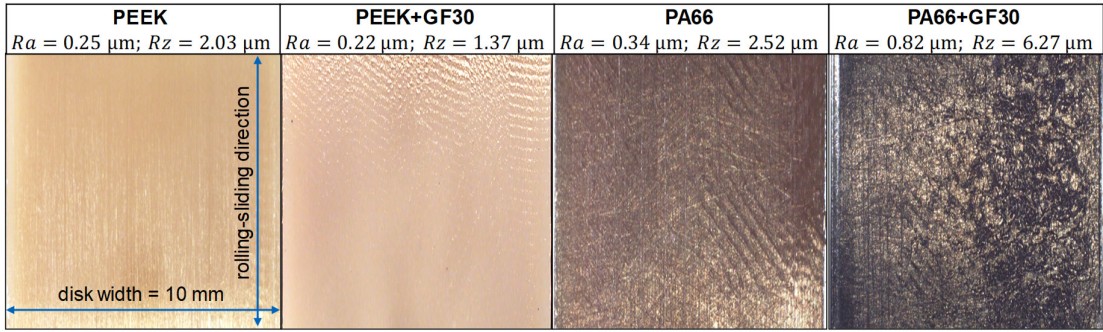

**Figure 2.** Surface impressions of all investigated polymer disks and averaged roughness values. PA66, polyamide 66; PEEK, polyetheretherketone; GF, glass fiber.

Analogously to many technical applications, the main setup is a hybrid polymer–steel pairing owing to its superior heat removal. A plain polymer pairing is used as a complementary setup. The lubricants are mineral oil (FVA3 [36]), a polyalphaolefine-based lubricant (PAO), and a water-containing lubricant (WAT). The key properties of each lubricant are summarized in Table 2. No surface-active additives are added.

**Table 2.** Properties of the used lubricants [36]. PAO, polyalphaolefine-based lubricant; WAT, water-containing lubricant; FVA3, mineral oil.

|  | FVA3 | PAO | WAT |
|---|---|---|---|
| Kinematic viscosity $v(40°C)$ $in$ $mm^2/s$ | 95 | 100 | 90 |
| Kinematic viscosity $v(60°C)$ $in$ $mm^2/s$ | 40 | 40 | 40 |
| Kinematic viscosity $v(100°C)$ $in$ $mm^2/s$ | 11 | 14 | 15 |
| Density $\rho$ $in$ $g/cm^3$ | 0.896 (15°C) | 0.894 (20°C) | 1.090 (20°C) |

*2.2. Operating Conditions and Experimental Procedure*

Table 3 shows the considered operating conditions, which are based on preliminary experiments and relevant for rolling-sliding conditions along the path of contact of polymer gears.

**Table 3.** Considered operating conditions and investigated influences at the FZG twin-disk test rig.

|  | Load sequences (temperature behavior) | Slip curves (friction and temperature behavior) |
|---|---|---|
| Normal force $F_N$ in N | > 1000 | **1000**; 1500 |
| Sum velocity $v_\Sigma$ in m/s | 1; 4; 16 | |
| Slip ratio s in % | 0 | 0/5/10/20/30/40/50/60/70 |
| Oil temperature $\vartheta_{oil}$ in °C | 60 | 40; **60**; 100 |
| Lubricants | FVA3 | **FVA3**; PAO; WAT |
| Materials | PA66; PA66+GF30; PEEK; PEEK+GF30 | |
| Setup of disks | polymer-steel | **polymer-steel** / polymer-polymer |

If Hertzian theory is applied for the sake of simplicity, a load of $F_N = 1000\ N$ results in Hertzian pressures $p_H$ of 74 N/mm² for PA66, 81 N/mm² for PEEK, 135 N/mm² for PA66+GF30, and 139 N/mm² for PEEK+GF30.

The load sequence experiments focus on the temperature behavior under increasing load for the hybrid TEHL contact to elaborate critical operating conditions and to avoid premature failures in the slip curve experiments. Therefore, under pure rolling conditions and constant sum velocity, the load is from $F_N = 1000\ N$ increased in steps of 200 N until the system cannot become quasi-stationary. The criterion for a quasi-stationary state is a bulk temperature change of less than $\Delta\vartheta_M \leq 0.5\ K/min$. In the case no quasi-stationary state is reached, after about 700 s, the load is increased. The load sequences were repeated at least twice for each material.

The slip curve experiments examine the friction and temperature behavior in the TEHL contact. On the basis of the load sequence experiments, a moderate load of $F_N = 1000\ N$ is set for the majority of experiments to avoid premature failure. Starting from pure rolling, the slip ratio s is increased in steps, as stated in Table 3, after each step has reached a quasi-stationary state ($\Delta\vartheta_M \leq 0.5\ K/min$) or after 700 s at the latest. The measuring points of the coefficient of friction and the bulk temperature shown in Section 3 are the average of measured values within one minute after reaching the quasi-stationary criterion. Owing to hydrodynamics, the lubrication conditions that occur are strongly influenced by the sum velocity. Thus, for example, the operating conditions shown in bold in Table 3 for PEEK result in relative lubricant film thicknesses ($\lambda_{rel} = h_m/(0.5 \cdot (Ra_1 + Ra_2))$ [37]) of $\lambda_{rel} \approx$ 0.75 for $v_\Sigma = 1\ m/s$ and of $\lambda_{rel} \approx 5.28$ for $v_\Sigma = 16\ m/s$. For PA66+GF30, the relative lubricant film thickness is $\lambda_{rel} \approx 0.33$ for $v_\Sigma = 1\ m/s$ and $\lambda_{rel} \approx 2.28$ for $v_\Sigma = 16\ m/s$.

According to that, $v_\Sigma = 1\ m/s$ can be assigned to the mixed lubrication regime, which is characterized by a high solid load portion. A sum velocity of $v_\Sigma = 16\ m/s$, on the other hand, can be associated with the fluid film lubrication regime characterized by the high fluid load portion.

Before each slip curve, the disks are thermally conditioned by operating the twin-disk test rig with separated disks at $v_\Sigma = 1\ m/s$ with activated oil injection. Thereby, a quasi-stationary initial bulk temperature of $\vartheta_{M,0}$ ($\vartheta_{oil} = 60\ °C$) $\approx 34\ °C$ is obtained. In general, a test cycle consists of the three different sum velocities shown in Table 3. Each sum velocity level is repeated three times with

the same specimen. No additional running-in process is performed. New test specimens are used each time after an influencing factor according to Sections 3.2.1 to 3.2.6 has been investigated.

## 3. Results and Discussion

In the following section, the results of the load sequences and slip curves investigations are presented and discussed. The shown error bars indicate the minimum and maximum measured values of each measuring point.

### 3.1. Temperature Behavior in Load Sequences

Figure 3 shows the measured averaged bulk temperature under pure rolling as a function of the normal force $F_N$ at different sum velocities $v_\Sigma$ in the hybrid polymer-steel setup. Besides the influence of an increasing load on the temperature behavior, the operating limits for subsequent damage-free slip curve investigations are determined. An essential indication for determining such limits is the development of the bulk temperature depending on the sum velocity and load, as the operating temperature of thermoplastics is generally limited owing to the strong temperature dependence of the material properties.

As illustrated in Figure 3, all materials show an increase in bulk temperature $\vartheta_M$ with increasing load at all sum velocities. For PEEK and PEEK+GF30, the increase is almost linear and significantly lower than for PA66 and PA66+GF30. The discrepancies in bulk temperature between the two thermoplastic types increase significantly with increasing sum velocity or rolling velocity. Unlike PA66, the bulk temperature difference between PEEK and fiber-reinforced PEEK is moderate. Especially, PA66 reaches remarkably and critically high bulk temperatures of $\vartheta_M > 100\,°C$ at a sum velocity of $v_\Sigma = 16\,m/s$ and a load of $F_N = 1400\,N$, which represents an increase in bulk temperature to PEEK of more than 40 K.

Owing to pure rolling conditions, the heating caused by interfacial friction are assumed to be very low. The energy losses referred to as solid losses, which are caused by the viscoelastic damping behavior of thermoplastics, are seen as the driving mechanism for the increase in bulk temperature. High solid losses for PA66, especially in the range of $\vartheta_M \approx 50 - 80\,°C$, are consistent with the loss modulus maximum in the range of the glass transition temperature $\vartheta_G$ (see Table 1). Bulk temperatures in the range of the glass transition temperature of PEEK and PEEK+GF30 are not reached in these experiments, which explains the comparatively low temperature response of both materials. At temperatures above the glass transition temperature, PA66 exhibits increasingly unstable material behavior. Owing to the high bulk temperatures, it is assumed that the proportion of plastic effects progressively increases. This also considers the fact that the Pt-100 sensors have regularly been damaged in this area, probably owing to a major deformation of the corresponding borehole.

The influence of the increased mechanical stiffness owing to fiber reinforcement is particularly evident at higher sum velocities. Solid losses are minimized owing to the lower strain in the material. A decisively enhanced thermal effusivity, also known as thermal inertia [38], by the fiber reinforcement cannot be assumed, as it has little effect on the thermal conductivity of the whole composite (see Table 1).

The load sequence experiments show the dominant influence of the viscoelastic material behavior on the heat balance in thermoplastic TEHL contacts. Especially in the range of the glass transition temperatures, disproportionate bulk temperatures arise for polyamide, which leads to the assumption that both plastic and viscoelastic material behavior may occur. For this reason, a conservative load of $F_N = 1000\,N$ is defined for the following investigations in Section 3.2, because even slightly higher loads for a sum velocity of $v_\Sigma = 16\,m/s$ lead to critical bulk temperatures, and cause unstable material behavior or failure.

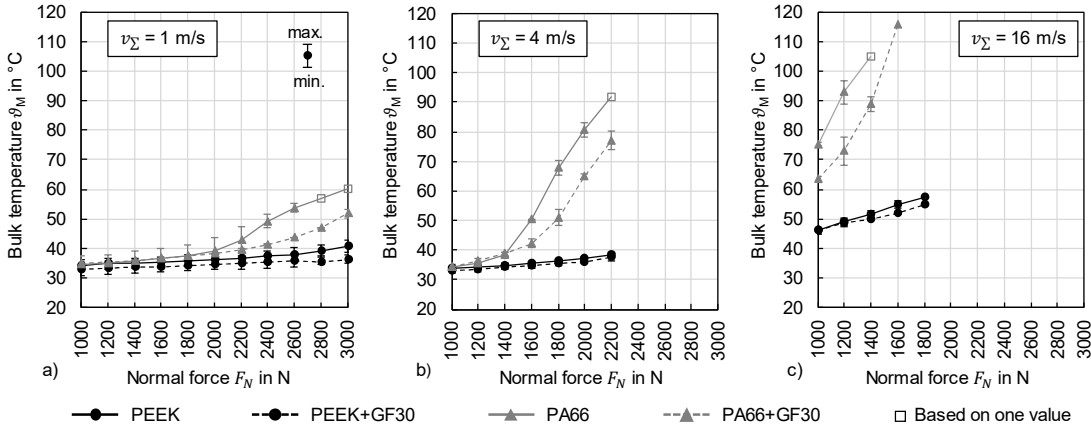

**Figure 3.** Bulk temperatures under pure rolling and increasing load for $v_\Sigma = 1\,m/s$ (**a**), $v_\Sigma = 4\,m/s$ (**b**), and $v_\Sigma = 16\,m/s$ (**c**) (hybrid polymer-steel setup; mineral oil (FVA3)).

### 3.2. Friction and Temperature Behavior in Slip Curves

On the basis of Section 3.1, the experimental results are presented by illustrating the coefficient of friction μ* and bulk temperatures $\vartheta_M$ over the slip ratio s. The measured coefficients of friction correspond mainly to the interfacial friction in the disk contact and are partially adapted according to the measurement principle of the test rig, as explained in Section 2.1.

#### 3.2.1. Influence of the Matrix Material

First, the influence of the two different matrix materials PA66 and PEEK on the friction and temperature behavior is considered for $F_N = 1000\,N$, $\vartheta_{oil} = 60\,°C$, and FVA3. In general, both materials have similar mechanical properties (see Table 1). However, there are differences in the temperature-dependent material behavior, especially the glass transition temperature $\vartheta_G$. If Hertzian theory is applied for the sake of simplicity, the contact pressure in the compliant PEEK–steel contact is $p_H \approx 81\,N/mm^2$, while it is $p_H \approx 73\,N/mm^2$ for the PA66–steel contact. Note that a plain steel contact results in $p_H \approx 429\,N/mm^2$ at the same normal force.

The experimental friction behavior measurements are shown in Figure 4. For conditions with high fluid load portion at $v_\Sigma = 16\,m/s$, very low coefficients of friction are measured because of the significant compliance of the surface, which keeps the effective viscosity, and thus the interfacial friction, low. The friction curve runs approximately linear, which indicates Newtonian lubricant rheology and confirms preliminary studies [6] (see Section 1). By decreasing the sum velocity to $v_\Sigma = 4\,m/s$ and $v_\Sigma = 1\,m/s$, the lubricant film thickness decreases and the solid load portion increases. For conditions with a high solid load portion at $v_\Sigma = 1\,m/s$, higher coefficients of friction are measured compared with $v_\Sigma = 16\,m/s$, even under pure rolling conditions (s = 0%). Depending on the material, temperature, and shear modulus, it is expected that the contact geometry will be affected and micro-slip will occur, thereby causing increased interfacial friction.

The bulk temperature behavior of the two matrix materials, however, differs significantly (Figure 5). Particularly at $v_\Sigma = 16\,m/s$ and even at pure rolling, a sharp increase in bulk temperature reflects losses in the polymer solid, despite the low measured friction, which mainly refers to interfacial friction. Exemplary for this, Figure 5a shows the bulk temperature curve of PEEK and PA66 under pure rolling as a function of time. Starting from a common initial bulk temperature of $\vartheta_{M,0} \approx 34\,°C$, the bulk temperature of PA66 rises by 35 K and by 15 K for PEEK in 700 s under identical operating conditions. This indicates the pronounced frequency-dependent behavior, particularly for PA66, resulting in increased solid losses. Basic dynamic material characterization assigns maximum solid losses to a specific frequency or temperature range that can be shifted, according to the time–temperature superposition principle. In contrast to PEEK, a critical frequency range for PA66 seems to be reached at $v_\Sigma = 16\,m/s$. Figure 5b shows quasi-stationary values for various sum velocities at

an increasing slip rate. For $v_\Sigma = 1\ m/s$, the bulk temperature remains almost constant with a slight increase originating probably from the interfacial friction heating. The steep bulk temperature rise by up to 35 K for PA66 at $v_\Sigma = 16\ m/s$ is the result of dominant solid losses, as described above. The negative gradient of the bulk temperature at $v_\Sigma = 16\ m/s$ with increasing slip ratio for PA66 is an indication of the strong frequency dependence on the damping behavior, as the surface velocity (frequency) of the polymer disk decreases (see Section 2.1). The heating by solid losses is less pronounced at lower sum velocities $v_\Sigma = 4\ m/s$ and $v_\Sigma = 1\ m/s$.

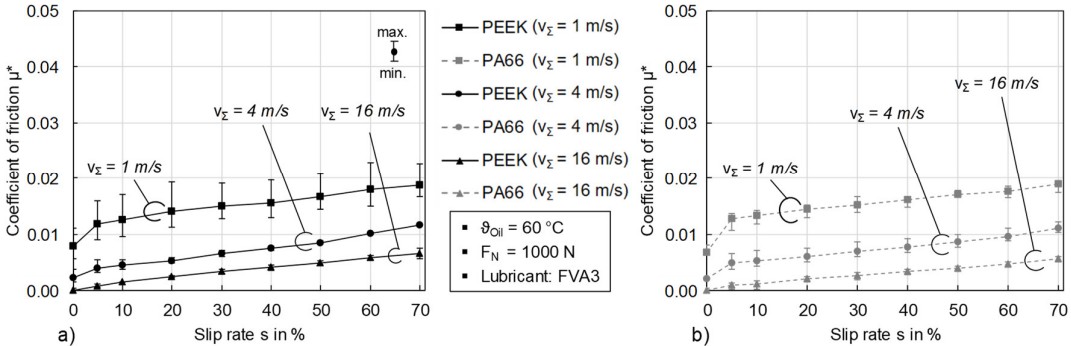

**Figure 4.** Friction behavior for increasing slip rates of PEEK (**a**) and PA66 (**b**).

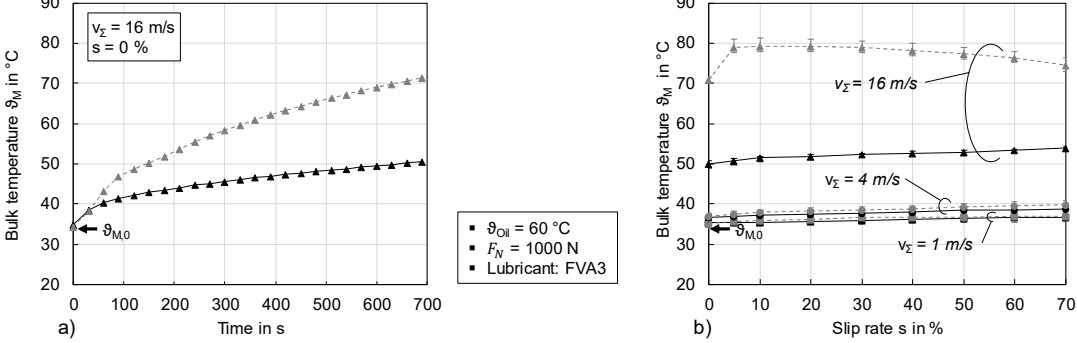

**Figure 5.** Temperature curve under pure rolling conditions (**a**) and temperature behavior for increasing slip rates (**b**) of PEEK and PA66.

3.2.2. Influence of the Lubricant

Figure 6 shows for PA66 and PEEK the coefficient of friction and bulk temperature curves in comparison with the three lubricants FVA3, PAO, and WAT under conditions with a high fluid load portion at $v_\Sigma = 16\ m/s$ (Figure 6a) as well as a high solid load portion at $v_\Sigma = 1\ m/s$ (Figure 6b) for $F_N = 1000\ N$ and $\vartheta_{oil} = 60\ °C$. The friction curves at $v_\Sigma = 16\ m/s$ in Figure 6a are almost identical in comparison with the lubricants and show a linear slope with increasing slip rate. The values of $\mu^* < 0.01$ correspond to an interfacial friction level of super lubricity. All lubricants show Newtonian rheology, while the base viscosity essentially determines the fluid friction. The friction curves at $v_\Sigma = 1\ m/s$ in Figure 6b remain at a low level for WAT, indicating low interfacial solid friction under these conditions. Note that the 20% higher density compared with PAO and FVA3 supports the lubricant film formation of WAT. Therefore, and likely owing to the higher interfacial solid friction, PAO and FVA3 show a higher coefficient of friction. The measured bulk temperature curves for $v_\Sigma = 1\ m/s$ and $v_\Sigma = 16\ m/s$ generally agree with the trends described in Section 3.2.1. In comparison with the lubricants, WAT tends to result in slightly lower bulk temperatures $\vartheta_M$ at $v_\Sigma = 16\ m/s$. This effect indicates increased heat dissipation supported by the water content in the lubricant owing to its favorable calorimetric properties. A comparable effect does not occur at $v_\Sigma = 1\ m/s$, as there is generally no significant increase in bulk temperature.

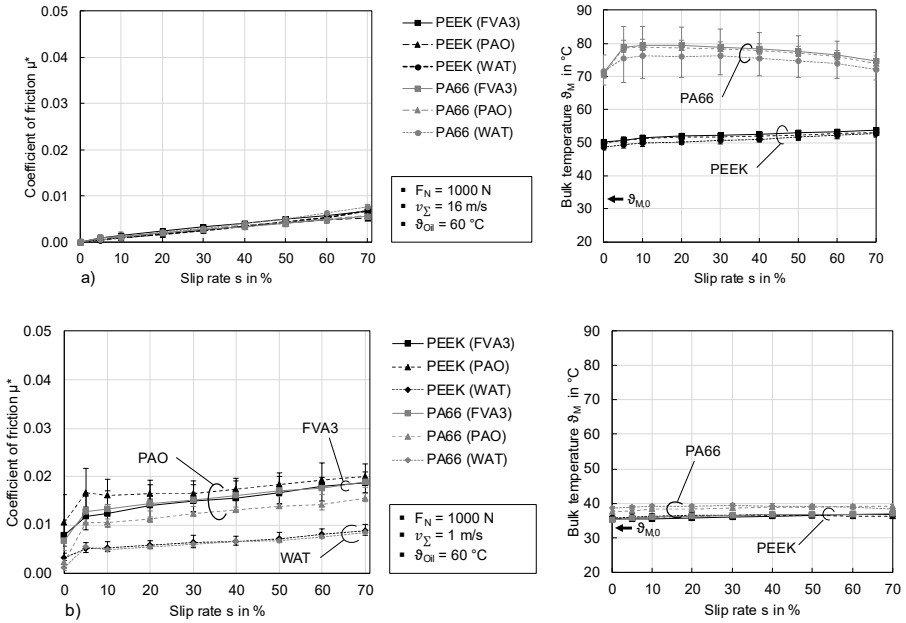

**Figure 6**. Influence of the lubricant on the friction and temperature behavior for PA66 and PEEK for $v_\Sigma = 1\ m/s$ (**a**) and $v_\Sigma = 16\ m/s$ (**b**).

### 3.2.3. Influence of the Oil Temperature

Figure 7 shows the influence of oil temperature on the friction and temperature behavior. Thereby, conditions of high fluid load portion ($v_\Sigma = 16\ m/s$) and high solid load portion ($v_\Sigma = 1\ m/s$) are considered at $F_N = 1000\ N$ for PEEK with FVA3 oil. With increasing oil temperature $\vartheta_{oil}$, the viscosity $v$ of FVA3 decreases significantly from $95\ cSt$ at $40\ °C$ to $40\ cSt$ at $60\ °C$ to $11\ cSt$ at $100\ °C$ (see Table 2). The friction curves at $v_\Sigma = 16\ m/s$ for the different oil temperatures are fairly close together. However, there are more pronounced effects on the frictional behavior under conditions with a high solid load portion at $v_\Sigma = 1\ m/s$. Here, the measured coefficient of friction increases with higher oil temperatures and lower viscosity, which indicates lower lubricant film thickness and increasing solid load portions. It is observed that a high solid load portion shows higher scattering in the measurements compared with $v_\Sigma = 16\ m/s$. The bulk temperature curves for $v_\Sigma = 16\ m/s$ agree with the measurement results in Section 3.2.1 and 3.2.2, characterized by solid losses. However, the shift of the temperature curves due to different oil temperatures $\vartheta_{oil}$ should be noted. All curves indicate that the damping behavior of PEEK does not change significantly in the considered oil temperature range, as the glass transition temperature is around $T_G \approx 143\ °C$ (see Table 1). Despite the comparably high measured coefficient of friction at $v_\Sigma = 1\ m/s$, the bulk temperature remains almost constant over the slip rate as well. This leads to the assumption that the interfacial frictional heat is insignificant and accumulates in the disk contact owing to the low thermal effusivity of polymers or is dissipated effectively by the lubricant and through the steel disk.

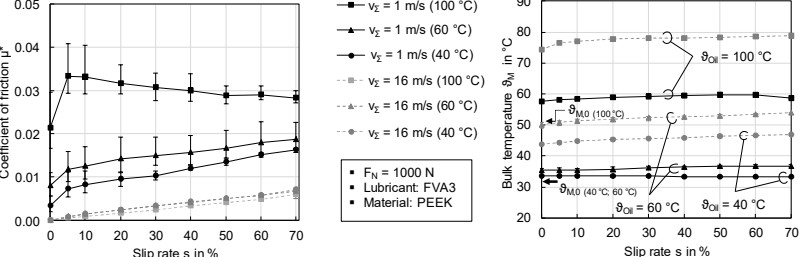

**Figure 7.** Influence of the oil temperature on the friction and temperature behavior for PEEK and FVA3.

### 3.2.4. Influence of the Load

Figure 8 shows the coefficient of friction and bulk temperature curves compared with $F_N = 1000\,N$ and $F_N = 1500\,N$ for $\vartheta_{oil} = 60\,°C$, PEEK, and FVA3. In contrast to lubricated steel contacts, the coefficient of friction remains almost constant or decreases with increasing load. By simply applying Hertzian theory, the corresponding increase in $F_N$ from $1000\,N$ to $1500\,N$ results in only moderate growth in contact pressure of about $\Delta p_H = 20\,N/mm^2$. The high compliance of the polymers favors the formation of surface conformity in the hybrid setup and keeps the pressure level low. For conditions with a high fluid load portion at $v_\Sigma = 16\,m/s$, the influence of the load increase on the lubricant film thickness and viscosity, and thus on fluid friction, is small. Assuming a constant Young's modulus and an oil temperature of $\vartheta_M \approx 60\,°C$, the minimum lubricant film thickness $h_m$ at $1000\,N/1500\,N$ is approximately $0.17\,\mu m/0.16\,\mu m$ and the corresponding kinematic viscosity $\nu$ is approximately $102\,mm^2/s/125\,mm^2/s$, which results only in a small change in the measurable interfacial friction force. Considering the definition of the coefficient of friction as $F_R/F_N$, this effect does not lead to an increase in the coefficient of friction. For conditions with a high solid load portion at $v_\Sigma = 1\,m/s$, the coefficient of friction even decreases with increasing load. It is assumed that, owing to higher contact conformity, in combination with higher solid losses and bulk temperatures at $1500\,N$ compared with $1000\,N$, the dominant interfacial solid friction decreases. This cannot be seen at $v_\Sigma = 16\,m/s$, as no solid load portion is expected.

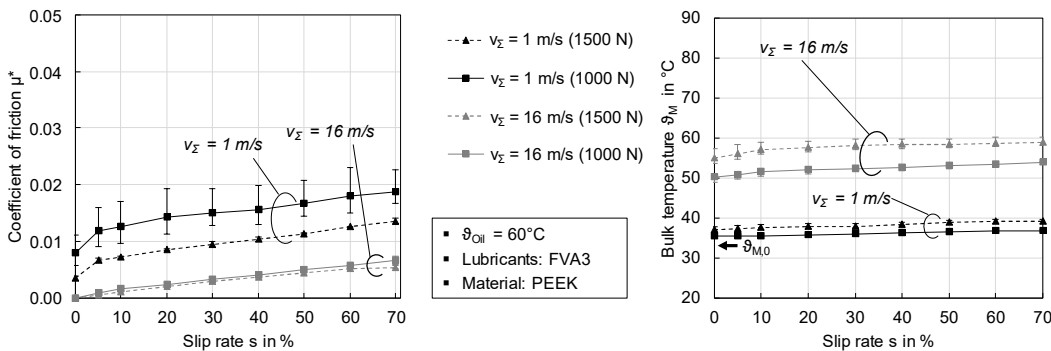

**Figure 8.** Influence of the load on the friction and temperature behavior for PEEK and FVA3.

### 3.2.5. Influence of the Disk Setup

Figure 9 compares the polymer–steel disk setup with the polymer–polymer disk setup at $F_N = 1000\,N$, $\vartheta_{oil} = 60\,°C$, and FVA3. The high fluid load portion at $v_\Sigma = 16\,m/s$ shows a slightly lower gradient of the friction curve for the PEEK–PEEK contact compared with the PEEK–steel contact. This behavior is connected to the low thermal effusivity of polymers and thermal insulation in the contact region. For the PEEK–PEEK contact, more heat is accumulated than for the PEEK–steel contact, which leads to a more pronounced decrease in viscosity and fluid friction. On the other hand, higher coefficients of friction for the PEEK–PEEK contact under conditions with a high solid load portion at $v_\Sigma = 1\,m/s$ indicate higher interfacial solid friction. The small differences in the coefficient of friction are attributed to the different surface structure of the polymer and steel specimens.

The bulk temperature measured 4 mm below the disk surface is $v_\Sigma = 16\,m/s$ lower for the PEEK–PEEK contact than for the PEEK–steel contact. Considering the effective stiffness of both contacts and the proportional deformation per body, the solid losses are smaller in plain polymer contacts, and hence the bulk temperature is lower. At $v_\Sigma = 1\,m/s$, both disk setups show an approximately equal low thermal response, which underlines the comparable and small viscoelastic losses at the corresponding loading frequency. Depending on the frequency-dependent solid losses and the low thermal effusivity, there are indications that decisive differences between contact and bulk temperature can occur. Compared with the hybrid contact with polymer–steel disk setup, even

higher contact temperatures with simultaneously lower bulk temperatures are possible for the polymer–polymer contact. This falls in line with the simulation results by Maier et al. [6].

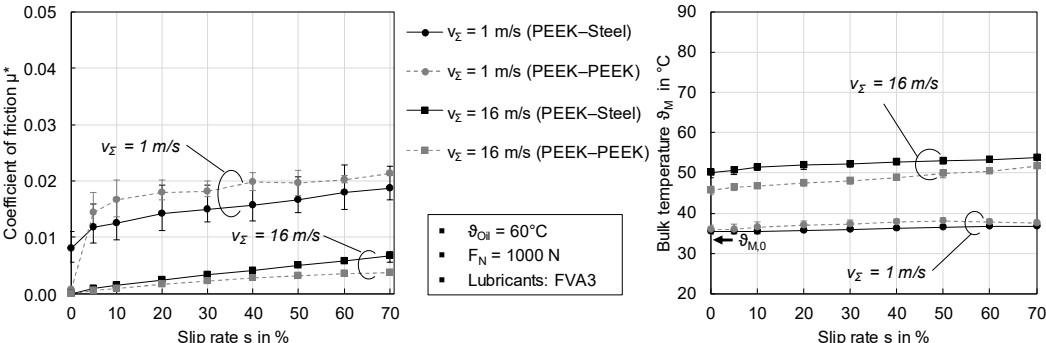

**Figure 9.** Influence of the different disk setups on the friction and temperature behavior for PEEK and FVA3.

### 3.2.6. Influence of Glass Fiber Reinforcement

Figure 10 shows for the hybrid polymer–steel disk setup the influence of glass fiber reinforcement on the friction and temperature behavior of PEEK and PA66 with $F_N = 1000\,N$, FVA3, and $\vartheta_M = 60\,°C$. For conditions with a high fluid load portion at $v_\Sigma = 16\,m/s$ in Figure 10a, PEEK+GF30 shows no significant difference in the friction behavior compared with plain PEEK. As the surfaces are well separated by a lubricant film, the viscosity of the lubricant mainly determines the interfacial friction. As surface asperities come into contact at $v_\Sigma = 1\,m/s$ with high solid load portion, PEEK+GF30 shows generally lower coefficients of friction. This is attributed to surface roughness, as PEEK+GF30 disk specimens are characterized by slightly lower Ra and Rz parameters compared with plain PEEK (see Figure 2). The surface of PA66+GF30 is characterized by a significantly rougher surface structure compared with the other materials, which is expressed in possible solid contacts even at $v_\Sigma = 16\,m/s$, causing higher coefficients of friction compared with plain PA66, as shown in Figure 10b. This underlines the influence of the surface structure by injection molding on the interfacial friction behavior, particularly at low sum velocities.

Considering the bulk temperature, there are no notable differences in the temperature behavior of PEEK and PEEK+GF30 for the operating conditions used in these investigations. However, for polyamide materials, different temperature curves are obtained. It is assumed that the higher stiffness and lower strain rates of PA66+GF30 compared with unreinforced PA66 lead to a reduction in solid losses. This is particularly evident in loss-critical temperatures around the glass transition point and at high loading frequencies. As solid losses are barely present at $v_\Sigma = 1\,m/s$, the slightly higher bulk temperature of PA66+GF30 must be associated with the surface structure and the high coefficient of friction.

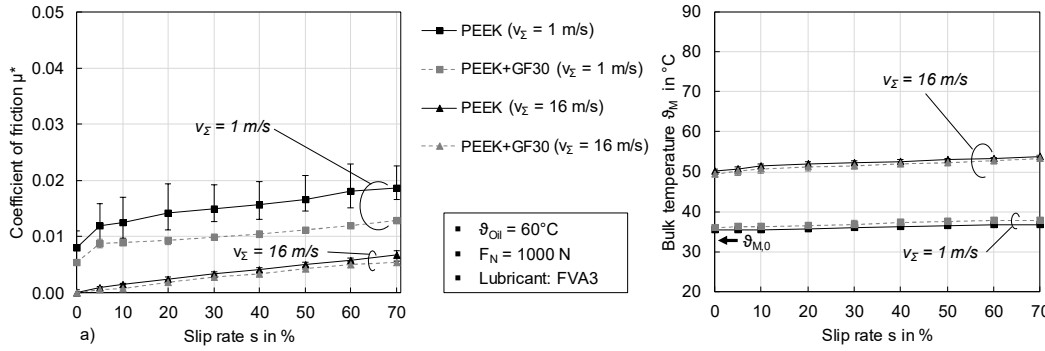

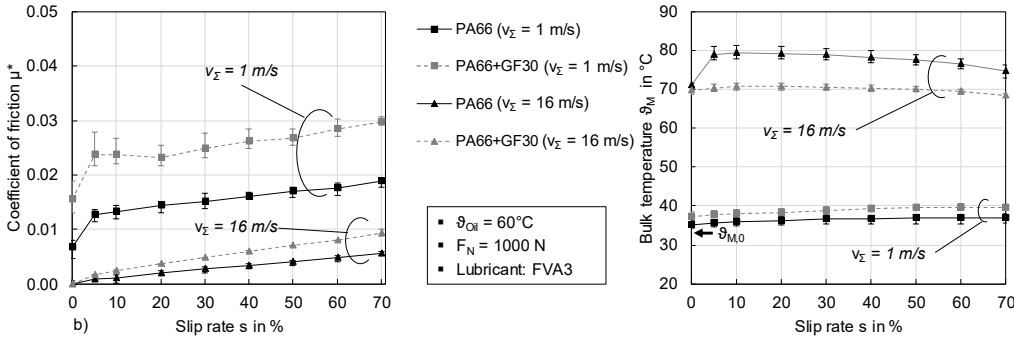

**Figure 10.** Influence of the 30 wt.-% glass fiber reinforcement on the friction and temperature behavior for PEEK (a) and PA66 (b).

### 3.2.7. Surface Impressions

An impression of the surfaces after the experimental investigations is given in Figure 11. It refers to representative surface images after slip curve experiments with $F_N = 1000\ N$, $\vartheta_{oil} = 60\ °C$, and FVA3. The images were generated using an optical microscope with 5x magnification. Compared with the initial condition (see Figure 2), small markers in the circumferential direction can be detected on the surfaces of PA66 and PA66+GF30. This indicates initial signs of wear. In contrast, the PEEK surface shows almost no visible changes or signs of wear. The indicated pattern, which displays the negative imprint of the injection mold, is visible on the surfaces. The exemplary roughness parameters for the PEEK/PEEK+GF30 confirm this assumption owing to almost constant values before and after the experiments (see Figure 11). In contrast, PA66/PA66+GF30 tends to show an increase in roughness, which follows the visual impressions.

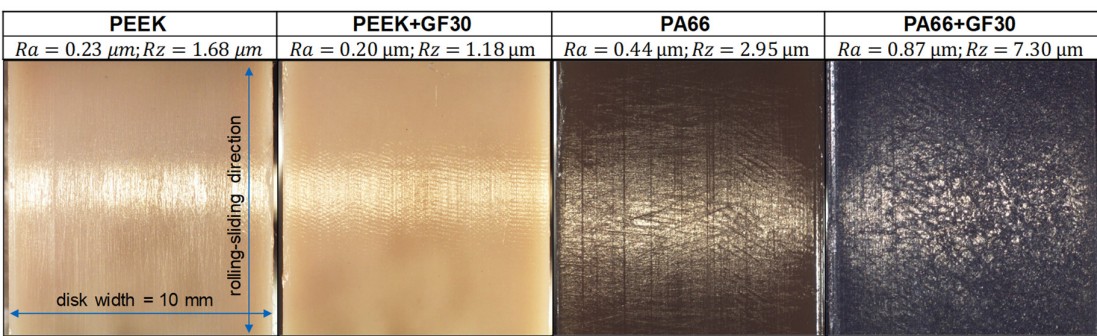

**Figure 11.** Impression of the surfaces after slip curve experiments for all materials ($F_N = 1000\ N$, $\vartheta_{oil} = 60\ °C$, FVA3, hybrid polymer-steel setup).

## 4. Conclusions

This work investigated the friction and temperature behavior of the thermoplastic TEHL contact in rolling-sliding conditions based on a phenomenological approach. Interfacial friction was found to be small even within the range of superlubricity owing to the low pressures that originate from high contact conformity. Under conditions with a high fluid load portion, lubricant viscosity determines the fluid's friction. The linear increase in the coefficient of friction with increasing slip ratio indicates Newtonian flow behavior for the considered lubricants. Under conditions with a high solid load portion, the interfacial friction is mainly influenced by the surface characteristic of the polymer such as roughness and the solid coefficient of friction. In this mixed lubrication regime, an increased lubricant viscosity reduces the solid contact ratio, and thus friction. Owing to the large surface deformations, increasing load causes only a small increase or even a decrease in the contact pressure and interfacial friction.

Despite the low interfacial friction, the viscoelastic material behavior of the polymer solids and the associated energy dissipation can lead to a significant increase in the bulk temperature. The solid losses in thermoplastic TEHL contacts depend mainly on loading amplitude and frequency. Compared with PEEK, PA66 shows more pronounced losses for the considered operating conditions, which relates to the glass transition temperature in combination with the considered loading frequencies. Especially past a critical frequency, which relates to the glass transition temperature by time–temperature superposition, the solid losses and bulk temperature strongly increase. In comparison, the heat generated owing to interfacial friction is considered low. An improvement of the frictional behavior by fiber reinforcement was not observed. However, for PA66, the solid losses of thermoplastics can be reduced by glass fibers and lower the bulk temperature, especially under critical conditions.

The considered twin-disk test rig with its force-based measurement principle has been frequently applied to plain steel contacts with loading-independent and mostly isotropic stiffness. The complex deformation behavior of polymer surfaces causes non-horizontal friction forces that are not registered by the load cell measuring friction force. For example, viscoelastic material behavior dampens the normal deformation, resulting in time-dependent normal stresses and solid losses. As such, the coefficients of friction presented here allow a direct comparison of variants with similar material deformation behavior, whereas the comparability of different polymer materials is limited. The measured bulk temperature is a good measure for the energy dissipation in solids, but is also superimposed by interfacial friction heat. Future loss torque measurements at the twin-disk test rig in addition to friction force measurement could allow the definition of a contact efficiency instead of a coefficient of friction, besides a breakdown of the origins of friction and losses in polymer contacts.

The range of applications for highly-stressed thermoplastic machine elements is continuously increasing. To enhance the transmittable power and prevent wear, lubricants are used more and more often. The phenomenologically gained experience on the friction and temperature behavior in this study can help to take advantage of thermoplastics available to technical applications and extend the range of applications towards higher power transmission.

**Author Contributions**: S.R. designed the experiments, analyzed the results, and wrote the paper. E.M. and T.L. supported the interpretation of the results, participated in the scientific discussions, and revised the paper. K.S. proofread the paper. All authors have read and agreed to the published version of the manuscript.

**Funding:** FZG would like to thank the German Research Foundation (DFG, Deutsche Forschungsgemeinschaft, STA 1198/15-1 // BO 1979/57-1) for its kind sponsorship of this research project focusing on the elastohydrodynamics of coated polymer gears. The presented results are also partly based on the research project IGF no. 18414 N/1 undertaken by the Research Association for Drive Technology e.V. (FVA); supported partly by the FVA and through the German Federation of Industrial Research Associations e.V. (AiF) in the framework of the Industrial Collective Research Programme (IGF) by the Federal Ministry for Economic Affairs and Energy (BMWi) based on a decision taken by the German Bundestag. The authors would like to thank for the sponsorship and support received from the FVA/AiF and the members of the project committee.

**Acknowledgments:** FZG would like to thank the Evonik Resource Efficiency GmbH (Kirschenallee; D-64293 Darmstadt) and the Teknor Germany GmbH (Am Rödlein 1: D-91541 Rothenburg ob der Tauber) for providing the raw material for the specimens. We also thank Klüber Lubrication München SE & Co. KG (Geisenhausenerstraße 7; D-81379 München) for providing the lubricants.

**Conflicts of Interest:** The authors declare no conflict of interest.

## Nomenclature

| | |
|---|---|
| E | Tensile modulus, N/mm$^2$ |
| $F_R$ | Friction force, N |
| $F_N$ | Normal force, N |
| $h_{min}$ | Minimal film thickness, µm |
| $L_t$ | Measuring distance of the roughness measurement, mm |
| PA66 | Polyamide 66 |
| PA66+GF30 | Polyamide 66 with 30 wt.-% glass fibers |

| PEEK | Polyetheretherketone |
|---|---|
| PEEK+GF30 | Polyetheretherketone with 30 wt.-% glass fibers |
| $\dot{Q}$ | Volume flow, l/min |
| $Ra$ | Arithmetic mean roughness, µm |
| $Rz$ | Maximum height, µm |
| s | Slip rate, % |
| $v_g$ | Sliding velocity, m/s |
| $v_\Sigma$ | Sum velocity, m/s |
| $v_1$ | Surface velocity (lower disk), m/s |
| $v_2$ | Surface velocity (upper disk), m/s |
| $\vartheta_G$ | Glass transition temperature, °C |
| $\vartheta_{M,0}$ | Initial bulk temperature, °C |
| $\vartheta_M$ | Bulk temperature, °C |
| $\vartheta_{oil}$ | Oil temperature, °C |
| $\mu^*$ | Coefficient of friction |
| $\nu$ | Oil kinematic viscosity, $mm^2/s$ |
| $\lambda_c$ | Cut-off wavelength, mm |
| $\lambda_{rel}$ | Relative film thickness |

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
