# Peer review of "Friction and Temperature Behavior of Lubricated Thermoplastic Polymer Contacts"

_lubricants, doi:10.3390/lubricants8060067_

Round 1
Reviewer 1 Report
The manuscript focuses on the friction and temperature behavior of soft TEHL contacts (PE and PEEK) under higher loads and rolling-sliding conditions using a two-disc setup. Thereby, authors discuss the influences of temperature, load, slip, lubricant, matrix material and glass fiber reinforcement.
The paper is well written and contains findings that justify publication. An sufficient overview of available literature is provided. The experimental methodology is appropriate and adequately described. Results are nicely presented and discussed. Therefore I recommend acceptance after the following points are adressed:
- How were the surface impressions recorded and the the roughness üaramters measured? How was this statistically confirmed? Since the authors also refer to roughness in their interpretation of the results in section 3.2.6 (which is basically understandable), further information should be given on this. In principle, 3D images, e.g. by WLI or LSM, would also be useful to substantiate the discussion.
- For the convenience of the reader, please add Hertzian parameters for the respective operating conditions already in section 2.2.
- The authors state that each setting was repeated 3 times? Was this done with new specimens?
Reviewer 2 Report
This paper reports a comprehensive study of the friction and temperature behavior of several thermoplastic polymer in rolling-sliding conditions with or without lubricants. The paper is very clearly written. The study is detailed with respect to understanding of the influences of thermoplastic material behavior on the tribological system. The results are interesting and exciting. Therefore, I recommend publication in Lubricants.
I have a few minor comments which the authors should address prior to publication:
1) All reagents used in experimental section should be accurately described, e.g. lack of resource and purity.
2) The authors mentioned that superlubricity can be achieved in their study due to low pressures that originate from high contact conformity. Several recent studies (1, 2) related to superlubricity due to other mechanisms (structural superlubricity, hydration lubrication, etc.) should be cited.
- Yaniv, R.; Koren, E. Robust Superlubricity of Gold–Graphite Heterointerfaces. Adv. Funct. Mater. 2019, 1901138.
- Lin, W.; Kampf, N.; Klein, J. Designer Nanoparticles as Robust Superlubrication Vectors. ACS Nano 2020, doi.org/10.1021/acsnano.0c01559.
